# Prevalence and pattern of adverse events following COVID-19 vaccination among adult population in Sokoto metropolis, northwest, Nigeria

Habibullah Adamu[1]*, Sufyanu Lawal[1], Ishaka Alhaji Bawa[1], Akilu Muhammad Sani[1], Adamu Ahmed Adamu[2]

1 Department of Community Health, Usmanu Danfodiyo University, Sokoto, Nigeria, 2 Department of Pharmacology and Therapeutics, Usmanu Danfodiyo University, Sokoto, Nigeria

* habibullah.adamu@udusok.edu.ng

## Abstract

### Background

COVID-19 still poses a major public health challenge worldwide and vaccination remains one of the major interventions to control the disease. Different types of vaccines approved by the World Health Organization (WHO) are currently in use across the world to protect against the disease. This study assessed the prevalence and pattern of adverse events following immunization (AEFI) after receiving COVID-19 vaccine (the Oxford-AstraZeneca vaccine) among the adult population in Sokoto metropolis, North-west, Nigeria.

### Methods

We conducted a cross-sectional study among 230 adults in Sokoto metropolis who received COVID-19 vaccines. Data was collected using a structured questionnaire administered via personal phone calls to respondents who were selected via a systematic sampling technique. For data analysis, IBM SPSS version 25.0 was used.

### Results

The majority of the participants [183 (79.7%)] experienced AEFI. The most common adverse events were body weakness [157 (85%)], fever [111 (60.3%)] and headache [103 (56%)]. Up to half of the respondents that experienced AEFI said it occurred within minutes and a few hours, whereas 75 (40.8%) said it was within 2-3 days. Up to 66.3% of the adverse reactions were mild and lasted between a few hours (37.5%) and one day (31.5%); however, 15.2% of the respondents had severe reactions of which 22.7% were admitted to a health facility. The development of AEFI was linked to the presence of an underlying medical condition (p = 0.001), a previous history of AEFI (p = 0.017), and a history of drug reaction (p = 0.005).

**Data availability statement:** Data cannot be shared publicly because of ethical confidentiality. Data are available from the researchers Ethics Committee (contact via the Principal researcher at: habibullah.adamu@udusok.edu.ng) for researchers who meet the criteria for access to confidential data.

**Funding:** The author(s) received no specific funding for this work.

**Competing interests:** The authors have declared that no competing interests exist.

## Conclusion

The majority of respondents reported adverse events following vaccination with the Oxford-AstraZeneca vaccine; body weakness, fever, and headache being the most common AEFIs. History of underlying medical condition as well as a history of adverse drug reactions were predictors of the development of adverse reactions following COVID-19 vaccination. Service providers at each COVID-19 vaccination point should always take the time to explain to vaccine recipients that adverse reactions are possible; however, they should reassure them that most ARs resolve within a few hours to a few days.

## Background

Severe acute respiratory syndrome coronavirus 2 (SARS-CoV-2) is the causative virus for the coronavirus disease 2019 (COVID-19) pandemic. SARS-CoV-2 first emerged in late 2019 in Wuhan (Hubei, China) and hastily became a global threat, affecting over 200 countries [1,2]. The pandemic has resulted in a devastating impact worldwide, which prompted the need for mitigation policies to contain it. Till date, there is no known cure for COVID-19, thus control measures are largely preventive in nature, and these include pharmacological and non-pharmaceutical interventions (NPIs) such as regular use of face masks, use of hand sanitizers, social distancing, travel restrictions, partial or complete lockdowns, and use of vaccines [2].

Because of the high disease burden of SARS-CoV-2, the development and manufacture of COVID-19 vaccines has been occurring at an unprecedented pace. Different vaccines against COVID-19 have now been approved for use by the World Health Organization (WHO); these vaccines include Oxford-AstraZeneca, Moderna, Pfizer-BioNtech, Johnson & Johnson and many others [3]. Different technologies have been adopted in the development of these vaccines and many are emerging with different mechanisms of action [3]. Findings from clinical trials have shown the vaccines to be effective and safe for use among humans [4]. The efficacy of Pfizer and Moderna vaccines in preventing disease or severe disease is 95–87.5% and 94.5–100%, respectively, whereas the efficacy of AstraZeneca and Janssen is about 70% and 65%, respectively [5]. Since the initiation of the COVID-19 vaccination programs in many countries, concerns have grown about adverse events following immunization (AEFI), especially with respect to COVID-19 vaccines [6,7].

An adverse event following immunization is any untoward medical occurrence which follows immunization and does not necessarily have a causal relationship with the use of the vaccine; the adverse event may be any unfavourable or unintended sign, abnormal laboratory finding, symptom or disease [8]. Because severe reactions following immunization are rare, several countries pool their AEFI data in a common global database; the database is managed by the WHO Programme for International Drug Monitoring. It has been observed that a significant number of severe AEFI are not true vaccine reactions but rather, coincidental occurrences of health events or the anxiety associated with the receipt of a vaccine [9].

Like with most vaccines, most local or systemic AEFI to COVID-19 vaccines are mild to moderate in severity, and the overall frequency of the AEFI varies with the age of the recipient and type of vaccine given, among others. Reports suggest that the rate of adverse reactions (ARs) to the Oxford-AstraZeneca vaccine is higher than that of the Pfizer-BioNtech vaccine. However, the ARs to the Oxford-AstraZeneca are less frequent in the older age group [10,11].

Several adverse events have been reported globally following the reception of different vaccines [12]. In a systematic review of studies conducted on the efficacy of COVID-19 vaccines across the globe, it was found that the proportion of individuals who had an adverse

reaction within 28 days after vaccination was lower than 30%. Most of the adverse reactions were mild to moderate and resolved within 24 hours after vaccination; pain and tenderness at the injection site were the most common local adverse reactions, whereas body weakness, pain and fever were the most frequent systemic adverse reaction [4]. In India, a study involving 1826 recipients of COVID-19 vaccines showed that 29.8% of them reported at least one of the AEFI; no severe adverse event was reported, and only 1.6% of them had moderate AEFI. Pain at the injection site (14.6%), fever (9.7%), and myalgia (5.9%) were the most common adverse events reported by the recipients. The AEFI incidence was higher in the first dose (38.1%) when compared to the second dose (26.4%) [13]. In Poland, a study among healthcare workers showed that 78.9% had at least one local reaction and 60.7% had at least one systemic reaction; pain at the injection site was the most prevalent local reaction (76.9%), while fatigue (46.2%), headache (37.7%) and muscle pain (31.6%) were the most common systemic reactions [14].

In Africa, an online survey conducted among participants from 22 African countries revealed that 80.8% of the respondents reported an adverse event following COVID-19 vaccination [12]. In Togo, at least one adverse event after receiving COVID-19 vaccine was reported among 71.6% of participants in a study; the most commonly reported adverse events were pain on injection site (91.0%), asthenia (74.3%), headache (68.7%), soreness (55.0%), and fever (47.5%). The prevalence of severe adverse events (SAEs) was 23.8%, with being a female under the age of 50 being more associated with the occurrence of SAEs [15].

In Nigeria, the pattern of adverse reactions to COVID-19 vaccines is similar to what has been observed in other countries; a cross-sectional study conducted in Rivers state showed that 50.5% of recipients of COVID-19 vaccine reported at least one adverse event post-vaccination, out of which 4.6% were said to be severe. The most common ARs were fever (73.0%), pain at the injection site (41.2%), fatigue (33.3%), body aches (17.5%) and headache (13.8%) [16]. In Sokoto state, up to 1,252,502 people have been fully vaccinated against COVID-19 [17].

Despite the number of people who have so far received the COVID-19 vaccine in Sokoto state, there is no available data on AEFI experienced by its recipients in the state. As fear of side effect has been mentioned as one of the reasons for COVID-19 vaccine hesitancy [18], this study therefore, aims to determine the prevalence and pattern of adverse reactions experienced by recipients of the vaccines in Sokoto state.

We hypothesized that, there is association between development of AEFI to COVID-19 vaccine and factors such as age, sex, history of underlying illness, history of AEFI to other vaccines, history of drug reactions/allergies, family history of AEFI/drug allergy etc.

## Materials and methods

### Study area

This study was carried out at Usmanu Danfodiyo University Teaching Hospital (UDUTH), Primary Healthcare Center (PHC) Arkilla and Specialist Hospital, all within the Sokoto metropolis.

UDUTH is a tertiary hospital that provides both health and academic services to Sokoto, Kebbi and Zamfara States and other parts of Nigeria, including some parts of neighbouring Niger and Benin Republics. It offers clinical services in the form of preventive, diagnostic, curative and rehabilitative services. The Centre for Advanced Medical Research and Training (CAMRET), which has been approved by the Nigeria Centre for Disease Control (NCDC) to carryout COVID-19 test is located within the hospital premises; UDUTH currently provides COVID-19 vaccination services using Pfizer BioNtech, Moderna, Oxford AstraZenica, and

Johnson and Johnson vaccines. Specialist Hospital Sokoto is a government owned general hospital located at Sultan Abubakar road, in Sokoto south LGA. The staff strength of over 1200 comprising doctors, nurses, pharmacists, medical laboratory scientists among others. The hospital is essentially a referral centre that offers preventive, curative and rehabilitative medical services. The hospital has specialty centres such as the Multidrug Resistant Tuberculosis Centre, Infectious Disease Clinic, Maternal and Child Health Clinic among others; it also provides COVID -19 vaccination services.

PHC Arkilla is a health facility located in Arkilla ward of Wamakko LGA, Sokoto state. It mainly provides primary health care services (preventive and curative) to the inhabitants of Arkilla ward, with an estimated population of about 30,000. PHC Arkilla has a staff strength of about 60 clinical and non-clinical staff comprising of Community Health Officers (CHOs), nurses/midwives, Community Health Extension Workers (CHEWs), Environmental Health Officers etc. The health facility also provides COVID-19 vaccination services.

## Study population

The study was conducted among the adult population who received COVID-19 vaccine within Sokoto metropolis.

## Inclusion criteria

Only individuals aged 18 years and above, who had received at least one dose of COVID-19 vaccine and had their names recorded in the COVID-19 registers of the selected health facilities were included in the study; both male and female recipients were included.

## Exclusion criteria

All recipients of COVID-19 vaccine whose phone numbers were not retrievable in the COVID-19 register at the respective hospitals were excluded.

## Study design and sampling

A cross-sectional study design was used; it was reported based on the STROBE (Strengthening the reporting of observational studies in epidemiology) guidelines/checklist for cross-sectional studies. The sample size was estimated using the Cochrane formula for estimating sample size in descriptive study [19].

$$N \;=\; Z^2 pq \,/\; d^2$$

where:

**n** = minimal sample desired in population greater than 10,000.

**Z** = standard normal deviate at 95% confidence interval = 1.96;

**p** = Prevalence of AEFI following Covid-19 vaccination from a previous study [12] = 80% = 0.80;

**q** = complimentary probability of p = 1-p; = 1 – 0.80 = 0.20

**d** = tolerable alpha error or level of precision = 5% = 0.05.

Therefore n =$1.96^2$x0.20x[1-0.20]/$0.05^2$

$$n = 245.90$$

Since the total population of those who received COVID-19 vaccine at the time of the study was less than 10,000, the sample size was adjusted using the formula for estimating sample size for a finite population ($n_f$) [19].

$$n_f = \frac{n}{1 + \frac{n}{N}}$$

$n_f$ = minimum sample size for a finite population

n = minimum sample size for infinite population

N = Total population of adults who received COVID-19 vaccine from the 3 selected health facilities (HFs) = 1300

$$\text{Therefore } n_f = \frac{245.8}{1 + \frac{245.8}{1300}}$$

$$N_f = 207$$

In cross-sectional studies, after a researcher has determined the minimum required sample size, it is important to provide additional allowances to cater for potential non-response subjects; this is because, not all study participants may likely complete the questionnaire. A minimum required sample size simply means the minimum number of subjects a study must have after recruitment is completed. To avoid underestimation of sample size, researchers will need to anticipate the problem of non-response and then to make up for it by recruiting more subjects on top of the minimum sample size. This is usually achieved by dividing the minimum sample size by the anticipated response rate. So if a researcher anticipates that up to 90% of the subjects will complete the questionnaires, he should divide the sample size by 90% (i.e., 0.9). Response rate is usually lower if the method of administering the questionnaire is by self-administration; thus more subjects will need to be recruited; in such situations, response rates as low as 60-70% could be anticipated [20,21].

Because the method of questionnaire administration adopted in our research was interviewer-administration (not self-administration), we anticipated a higher response rate (90%), thus we adjusted for non-response by dividing the calculated minimum sample size (207) by the 90% anticipated response rate (i.e., 0.9) [20,21].

=207/0.9

=230

therefore, 230 study participants were enrolled into the study.

## Sampling technique

UDUTH, Specialist Hospital Sokoto and PHC Arkilla were selected out of the health facilities that provide COVID-19 vaccination services in Sokoto metropolis using simple random sampling technique. From each of the three selected HFs, systematic sampling technique was used to select study participants; the COVID-19 register in each of the vaccination points was used as sampling frame. Sampling interval for selection of participants in each HF was calculated by dividing the total number of COVID-19 vaccine recipients in the register by the sample size for that HF.

## Instrument for data collection

Data was collected using a set of interviewer administered questionnaire which was prepared in English language and translated to the local language–Hausa through a two-way process to verify the accuracy of the translation by two Hausa scholars. The questionnaire had three sections as follows:

Section A: Sociodemographic profile of respondents

Section B: Prevalence and pattern of AEFI following COVID-19 vaccination

Section C: Factors associated with adverse reactions following COVID-19 vaccination

## Personnel

The researchers trained three research assistants who assisted in the data collection. They were trained for two days; each training session lasted for 2 hours. The training on how to administer the questionnaire via phone calls, an overview of COVID-19 and COVID-19 vaccines, general principles of research, objectives of the study, conduct of research and interpersonal communication skills.

## Questionnaire administration and pretest

We pretested the questionnaire among adult recipients of COVID-19 vaccines in PHC Yar Akija. PHC Yar Akija is located in Sokoto north LGA and it provides preventive and curative primary health care services (including COVID -19 vaccination services) to a population of about 25,000. Following the pretest, a few amendments were made to the questionnaire. Internal consistency of the questionnaire was assessed using Cronbach's alpha; the questionnaire had Cronbach's alpha coefficient of 0.71, which was considered valid. The questionnaire was designed and deployed to a web-based account created on https://kobo.humanitarianresponse.info [22]. The deployed questionnaire was then accessed by downloading Open Data Kit (ODK) app on Android devices and used for data collection. The use of ODK App in data collection helps to prevent or minimize data entry errors, ease timely data collection, ensure completeness of the information and subsequent processing and analysis [22].

Data was collected by administering the questionnaire via personal phone calls (made by the trained research assistants) to all the selected participants whose phone numbers could be retrieved from the COVID-19 register. Before administering each questionnaire, the research assistants explained the purpose of the research to the respondent and sought their consent to participate. In this study, an adverse event following immunization was considered as any untoward medical occurrence after receiving COVID 19 vaccine; it included any unfavourable or unintended sign, abnormal laboratory finding, symptom or disease.[8]The overall process of recruitment of study participants and collection of data lasted for about 7-8 weeks (between March and May 2022).

## Statistical analyses

Data was retrieved from the ODK server, exported to Microsoft Excel 2016 and then transferred to IBM SPSS version 23 software for analysis. Categorical variables were analyzed and presented as frequencies and percentages, while quantitative variables were analyzed and presented as summary measures in form of measures of central tendency and their corresponding measures of dispersion. The Pearson chi-square test was used to determine the factors associated with adverse reactions following COVID-19 vaccination. A binary logistic regression model was used to identify factors that predicted AEFI following COVID-19 vaccination. The major outcome variable was the occurrence of AEFI following COVID-19 vaccination and the independent variables included sociodemographic characteristics of the study participants, previous history of AEFI, underlying medical illness and previous history of drug reactions, among others. The Level of statistical significance for all inferential statistical analysis was set at $p < 0.05$.

### Ethical approval for the study

Approval to conduct the study was obtained from the Health Research Ethics Committees of the Ministry of Health, Sokoto State, Nigeria (No. SKHREC/0115/2021) and Usmanu Danfodiyo University Teaching Hospital, Sokoto (No. HREC/2022/1128/V1). Prior to recruitment, verbal informed consent of all study participants was sought. Before obtaining the verbal consent of each participant, there was brief description of the research process including its aim and what was required from the participant; the study participants were also given assurance that the information they provided would be handled with utmost confidentiality.

## Results

A total of 230 questionnaires were administered, and all 230 were completed, retrieved, and analyzed, yielding a response rate of 100%. This response rate was achieved because the interviews were conducted at each study participant's convenient time; thus, several phone calls were made to the study participants in order to complete the questionnaires.

Majority of respondents were in the age group of 20-39 years of age (55.8%) with others between 40-59 years of age (33.8%). Most of the respondents were males (60.2%), married (75.8%) and practiced Islam as religion (95.7%) (Table 1).

**Table 1. Socio demographic profile of the respondents.**

| Variable | Frequency (n = 230) | Percentage (%) |
|---|---|---|
| **Age group (years)** | | |
| ≤20 | 2 | 0.9 |
| 20-39 | 129 | 55.8 |
| 40-59 | 78 | 33.8 |
| 60 and above | 21 | 9.5 |
| **Sex** | | |
| Male | 138 | 60.2 |
| Female | 92 | 39.8 |
| **Religion** | | |
| Christianity | 10 | 4.3 |
| Islam | 220 | 95.7 |
| **Tribe** | | |
| Hausa/Fulani | 215 | 93.1 |
| Igbo | 8 | 3.9 |
| Yoruba | 7 | 3.0 |
| **Marital status** | | |
| Married | 175 | 75.8 |
| Divorced | 7 | 3.0 |
| Widow | 5 | 2.2 |
| Single | 43 | 19.0 |
| **Occupation** | | |
| Health worker | 31 | 13.4 |
| Businessman/businesswoman | 101 | 43.7 |
| Self employed | 51 | 22.5 |
| Farmer | 8 | 3.5 |
| Student | 39 | 16.9 |

Majority of the participants [184 (79.7%)] have experienced AEFI following COVID-19 vaccination (Fig 1).

The most frequent AEFI experienced by the respondents was body weakness (85.0%), followed by fever (60.3%); the least AEFIs experienced by the respondent were rash (2.7%) (Fig 2).

Up to half of the respondents that experienced AEFI experienced it immediately (50.5%), whereas 40.8% experienced it within 2-3 days. Most of the reactions lasted between a few hours (37.5%) and one day (35.1%). About two-thirds (66.3%) of the reactions were mild and 15.2% were severe reactions, of which 22.7% required admission to a health facility (Table 2).

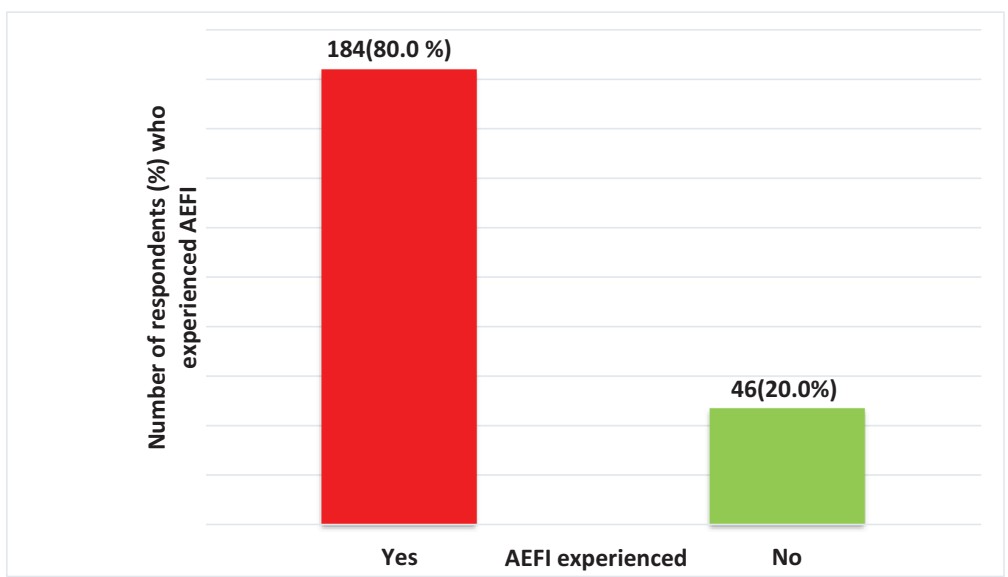

**Fig 1. Prevalence of AEFI among respondents.**

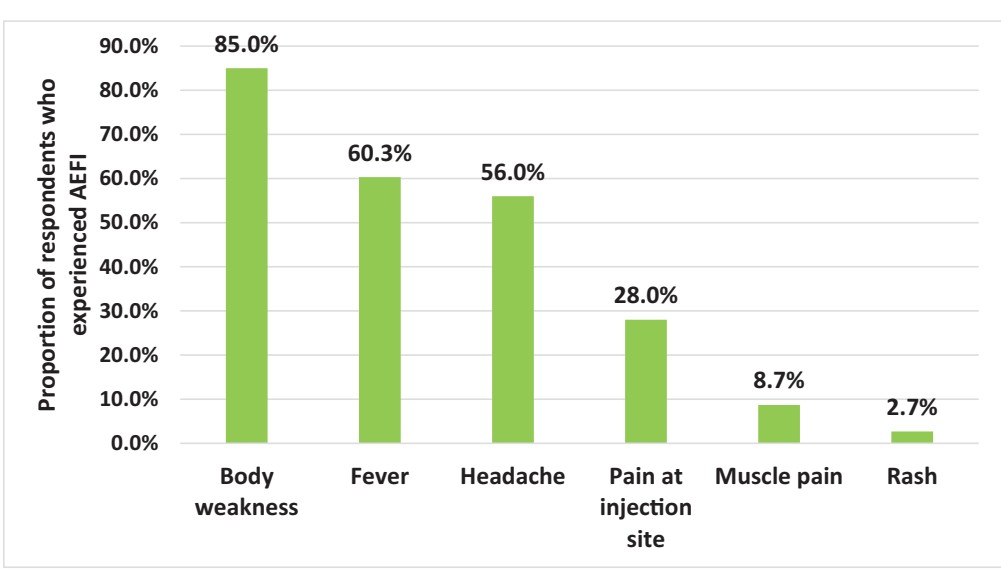

**Fig 2. Types of AEFI experienced by the respondents (Multiple options).**

Table 2. Pattern of AEFIs experienced by the respondents.

| Variable | Frequency (n = 184) | Percentage |
|---|---|---|
| **When did you experience AEFI?** | | |
| Immediately | 93 | 50.5 |
| Within 2-3 days | 75 | 40.8 |
| After 1 week | 10 | 5.4 |
| After 1 month | 6 | 3.3 |
| **For how long did you experience the reaction?** | | |
| Few minutes | 19 | 10.3 |
| Few hours | 69 | 37.5 |
| 1 day | 58 | 31.5 |
| 2 days | 37 | 20.1 |
| Others | 1 | 0.5 |
| **How severe was the reaction?** | | |
| Mild | 122 | 66.3 |
| Moderate | 34 | 18.5 |
| Severe | 28 | 15.2 |
| **If severe, were you admitted in a health facility as a result of the AEFI?** | | |
| Yes | 19 | 22.7 |
| No | 9 | 77.3 |

*Classification into mild, moderate, severe was based on respondents' perception of the AEFI.

Close to half of the respondents [88 (47.6%)] said they did nothing following their experience of AEF; 81 (44%) said they took some drugs; and 42 (22.6%) said they notified the vaccination about the adverse reaction (Fig 3).

In Table 3, factors found to be significantly associated with AEFIs to COVID-19 vaccine were history of an underlying medical illness (p = 0.001), a previous history of AEFIs to other vaccines (p = 0.019), and a previous history of drug reaction (p = 0.006). Other factors, such as a history of other allergies (other than drugs), cigarette and alcohol use, were not associated with AEFI to COVID-19 vaccines (p > 0.05).

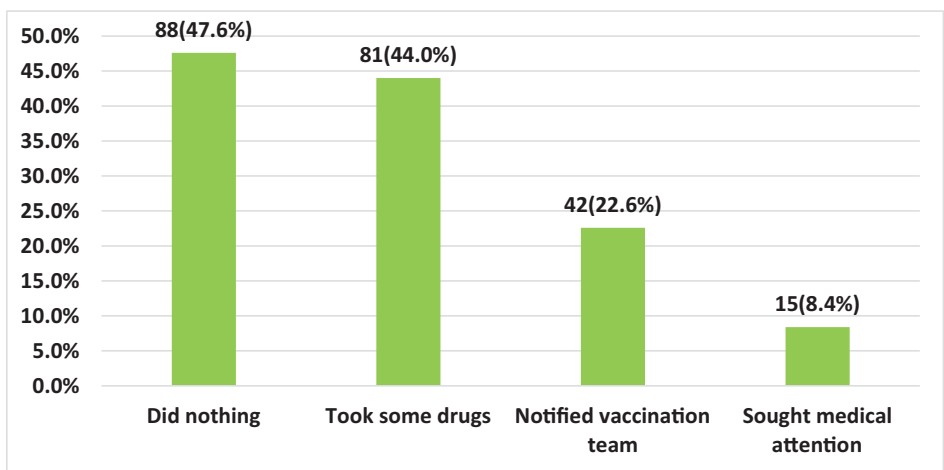

Fig 3. Actions taken after experiencing the AEFI (multiple options).

Table 3. Factors associated with AEFI following COVID-19 vaccination.

| Variable | AEFI Experienced | | Test statistic |
|---|---|---|---|
| | YES (%) | NO (%) | |
| **History of underlying illness** | | | |
| Yes | 61 (93.8) | 4 (6.2) | $\chi^2 = 10.857$ |
| No | 123 (74.5) | 42 (25.5) | **p = 0.001** |
| **Previous history of AEFI to other vaccines** | | | |
| Yes | 20 (100) | 0 (0.0) | $\chi^2 = 5.476$ |
| No | 164 (78.1) | 46 (21.9) | **p = 0.019** |
| **History of drug reaction in the past** | | | |
| yes | 34 (97.1) | 1 (2.9) | $\chi^2 = 7.582$ |
| No | 150 (76.9) | 45 (23.1) | **p = 0.006** |
| **Previous history of allergy to anything other than drug or vaccine** | | | |
| Yes | 15 (93.8) | 1 (6.3) | $\chi^2 = 2.032$ |
| No | 169 (79.0) | 45 (21.0) | p = 0.154 |
| **Family history of reactions following vaccination** | | | |
| Yes | 16 (94.1) | 1 (5.9) | $\chi^2 = 2.287$ |
| No | 168 (78.9) | 45 (21.1) | p = 0.130 |
| **Family history of drug allergy** | | | |
| Yes | 14 (93.3) | 1 (6.7) | $\chi^2 = 1.783$ |
| No | 170 (79.1) | 45 (20.9) | p = 0.182 |
| **History of cigarette smoking** | | | |
| Yes | 16 (94.1) | 1 (5.9) | $\chi^2 = 2.330$ |
| No | 166 (78.7) | 45 (21.3) | p = 0.127 |
| **History of alcohol consumption** | | | |
| Yes | 2 (100) | 0 (0.0) | $\chi^2 = 0.504$ |
| No | 182 (79.8) | 46 (20.2) | p = 0.478 |

$\chi^2$ = Pearson Chi square; Significant at p < 0.05.

Table 4. Binary logistic regression showing the predictors of AEFI following COVID-19 vaccination.

| Predictors | P-value | OR | 95% CI | |
|---|---|---|---|---|
| | | | Lower | Upper |
| **History of underlying medical condition** Yes vs. No[*] | 0.007 | 4.448 | 1.513 | 13.077 |
| **History of drug reaction in the past** Yes vs. No[*] | 0.023 | 10.427 | 1.389 | 78.272 |

[*]Reference category.

The binary logistic regression table (Table 4) shows that those who had a history of an underlying medical condition were about 4 times more likely to have AEFI following COVID-19 vaccination (p = 0.007, OR = 4.448, 95% C I = 1.513-13.077); likewise, those with a previous history of drug reaction were about 10 times more likely to experience the AEFI (p = 0.023, OR = 10.427, 95% CI = 1.389-78.272).

## Discussion

This study was conducted to determine the prevalence and pattern of AEFI following COVID-19 vaccination among the adult population in Sokoto metropolis North West, Nigeria.

In this study, up to 79.7% of the respondents have experienced AEFIs following COVID-19 vaccination and this is quite high especially for a vaccine that has recently been introduced. Similar studies conducted in Europe [14,23], Asia [11], North America [24] and Africa [12,15] also reported a high prevalence of adverse reactions to COVID-19. The high prevalence of the adverse reactions observed in these studies is not surprising considering the fact that all vaccines, including the vaccines used in the Expanded Program for Immunization (EPI), have varying forms of adverse reactions [9,25]. However, the adverse reactions are usually mild [25].

Up to two-thirds of the adverse reactions experienced by the respondents in this study were mild, which is also the case for most vaccines [9,25]. The adverse reactions reported by study participants (body weakness, fever, headache, and pain at injection site) are consistent with previous findings [13,15,16,26,27]. This pattern and types of adverse reactions have been observed to be more common among recipients of the AstraZeneca vaccine [10,11,26]; similarly, in this study, all the study participants received only the AstraZeneca vaccine. The side effects of most COVID-19 vaccines usually disappear within a few days of receiving the vaccine [11,23]; in this study as well, up to 99% of the side effects experienced disappeared within few a minutes to two days. As these side effects are usually mild and quite frequent, it is very important for service providers at the vaccination points to always take time to explain to the recipients of the vaccines that these adverse events may likely occur and that they should not be worried as most of it will resolve within few hours to few days. Providing recipients of vaccine with adequate information regarding the vaccine's side effects will help in alleviating some of the fears they have and by extension, this will improve compliance. It is however, important to explain that moderate to severe side effects can also occur [11,15,28] and in such cases, recipients are usually advised to report to the health facility for further evaluation; in this study, 15.2% of the respondents said they experienced severe adverse events. In Nigeria, the National Agency for Food and Drugs Administration and Control (NAFDAC) has designed a form (ADR Form) that is used in pharmacovigilance of drugs and food including cosmetics. This form is available in all the NAFDAC state offices in the 36 states of Nigeria, including the Federal Capital Territory (Abuja); the form is used for reporting adverse drug reactions (ADR). Adverse reactions to COVID 19 vaccines are also reported using this form, however, a major challenge faced in the use of this form for the pharmacovigilance of COVID 19 vaccines is the poor reporting of adverse reactions experienced by the recipients of the vaccine.

It is expected that recipients of the vaccines would report any adverse event experienced after receiving the vaccine; in this study, up to three-quarters of those that experienced an adverse reaction did not report it to the vaccination team. The poor reporting rate observed in this study could have resulted from the fact that the majority of the adverse events experienced by the respondents were mild, so the likelihood of a worsening to severe reaction was perceived to be low. When adverse events are not reported to the appropriate authorities, it makes it more difficult to understand how safe a given vaccine is, unless studies such as this are undertaken. The frequencies of adverse reaction occurring after COVID-19 vaccination differ according to the specific type of vaccine (AstraZenica, BioNtech, Morderna etc) received [9,14,29]; in this study we could not compare the adverse reactions experienced based on type of vaccine received because as at the time of collecting data for this study, the only vaccine available in all the three health facilities surveyed was the Astrezenica vaccine; therefore, all the study participants received only the Astrazeneca COVID-19 vaccine. Different studies conducted in different parts of the globe have reported numerically more incidences of AEs in recipients of the AstraZenica vaccine compared with those who received the BioNtech vaccine [30–32].

On bivariate analysis of factors associated with AEFI following COVID-19 vaccination, this study revealed that the presence of underlying medical conditions, previous history of reactions to other vaccines and history of drug reactions were statistically significantly associated with development of AEFI following COVID-19 vaccination. Herve' et al., also observed that, vaccine adverse effects are closely related to sex, age, underlying disorders, and drug history [33]. On multivariate analysis, however, two factors were found to be significant predictors of AEFI. Those who had a history of an underlying illness were about four times more likely to have AEFI after receiving the COVID-19 vaccine; likewise, those with previous history of drug reactions were about 10 times more likely to experience the AEFI. Similar observation was made in a studies conducted by Zare et al. [26], and Jamshidi et al. [34], which observed that, prevalence of adverse reactions following COVID-19 vaccination was associated with history of underlying illness. A study conducted in Vietnam to assess the factors influencing AEFI following COVID-19 vaccination with Oxford AstraZeneca among adult population also reported that underlying medical conditions and previous history of reaction to other vaccines were found to be significantly associated with AEFIs following COVID19 vaccination [35].

## Study limitation

One major limitation of this study is the fact that all the respondents received only the AstraZeneca vaccine, thus it was not possible to compare the rate of adverse events across all the vaccines approved for the COVID-19 vaccination. We also relied on respondents' self-reported adverse reactions, thus some adverse reactions could have been missed probably because the respondents forgot to mention them. Moreover, some of the respondents were interviewed several weeks after receiving the vaccine. Therefore, their responses could have been affected by some recall bias issues. Nevertheless, we believe their recall was good enough since all the adverse reactions were mentioned to the respondents during the phone call interviews.

## Study strength

We minimized sampling/selection bias by using a probability sampling technique (systematic sampling technique) to select the study participants, therefore each recipient of the vaccine was given equal chance of being selected. The COVID-19 register that was used as the sampling frame was up-to-date at the time of the data collection, thus no recipient of the COVID-19 vaccine was missed in the sampling process. In the methodology we adopted, we mentioned all the adverse reactions to the study participants, during the phone call interview; we believe this has significantly reduced the chances of recall bias. The data analyst, who is also one of the co-authors, did not participate in the data collection; univariate, bivariate and multivariate analyses were conducted on the data. We believe this has greatly reduced the effect of measurement bias.

## Conclusion

This study revealed that majority of the respondents experienced adverse events following vaccination with the Oxford-AstraZeneca vaccine and the most common AEFIs experienced by respondents were body weakness, fever and headache. The only predictors of adverse reaction following COVID-19 vaccination were the underlying medical condition and a history of drug reaction.

The study findings imply that it is very important for service providers to rule out any underlying medical illness and previous drug reaction before administering COVID-19 vaccine, as these were found to significantly predict adverse reaction to COVID-19 vaccine.

Now that there are up to four different COVID-19 vaccines in use in Sokoto state, there is need for further research to study the different patterns of the adverse reactions for each of the available COVID -19 vaccines. Given that this study only focused on recipients within the metropolis, there is also a need for similar research to study the pattern of AEFIs across the whole state.

## Supporting information

**S1 File. Covid-19 AEFI project GROUP 23 Clean data anonimized.**
(XLSX)

**S1 Questionnaire. Questionnaire on prevalence and pattern of adverse events following COVID-19 vaccination among adult population in selected health facilities in Sokoto Metropolis, North-West, Nigeria.**
(DOCX)

**S1 Checklist. STROBE checklist summary.**
(DOC)

## Author contributions

**Conceptualization:** Habibullah Adamu.

**Data curation:** Habibullah Adamu, Sufyanu Lawal, Ishaka Alhaji Bawa, Akilu Muhammad Sani.

**Formal analysis:** Habibullah Adamu, Sufyanu Lawal, Ishaka Alhaji Bawa.

**Funding acquisition:** Habibullah Adamu, Sufyanu Lawal, Ishaka Alhaji Bawa, Akilu Muhammad Sani, Adamu Ahmed Adamu.

**Investigation:** Habibullah Adamu.

**Methodology:** Habibullah Adamu, Akilu Muhammad Sani, Adamu Ahmed Adamu.

**Project administration:** Habibullah Adamu, Sufyanu Lawal.

**Resources:** Habibullah Adamu.

**Software:** Habibullah Adamu.

**Supervision:** Habibullah Adamu, Adamu Ahmed Adamu.

**Validation:** Habibullah Adamu, Ishaka Alhaji Bawa, Adamu Ahmed Adamu.

**Visualization:** Habibullah Adamu.

**Writing – original draft:** Sufyanu Lawal, Ishaka Alhaji Bawa, Akilu Muhammad Sani.

**Writing – review & editing:** Habibullah Adamu, Adamu Ahmed Adamu.

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
