## [Decision Letter · Decision Letter 0]

20 Mar 2023

Dear Dr. Adamu,

Thank you for submitting your manuscript to PLOS ONE. After careful consideration, we feel that it has merit but does not fully meet PLOS ONE’s publication criteria as it currently stands. Therefore, we invite you to submit a revised version of the manuscript that addresses the points raised during the review process.

2. The conclusion should based on the aim and objectives of the research

3. There also are errors in the references that should be corrected.

4. Other comments are in the manuscript.

We look forward to receiving your revised manuscript.

Kind regards,

Bilal Sulaiman

Academic Editor

PLOS ONE

Journal Requirements:

2. In the ethics statement in the Methods, you have specified that verbal consent was obtained. Please provide additional details regarding how this consent was documented and witnessed, and state whether this was approved by the IRB

Additional Editor Comments:

1. The reviewer has made comments that need to be addressed.

2. The conclusion should based on the aim and objectives of the research

3. There also are errors in the references that should be corrected.

4. Other comments are in the manuscript.

Reviewers' comments:

Reviewer's Responses to Questions

**Comments to the Author**

1. Is the manuscript technically sound, and do the data support the conclusions?

Reviewer #1: Yes

2. Has the statistical analysis been performed appropriately and rigorously?

Reviewer #1: Yes

3. Have the authors made all data underlying the findings in their manuscript fully available?

Reviewer #1: No

4. Is the manuscript presented in an intelligible fashion and written in standard English?

Reviewer #1: Yes

Reviewer #1: Overall comment:

The manuscript entitled “Prevalence and pattern of adverse events following COVID 19 vaccination among adult population in Sokoto metropolis, northwest, Nigeria” evaluated the prevalence and pattern of adverse events following immunization (AEFI) after receiving COVID-19 vaccine among the adult population in Sokoto metropolis, North-west, Nigeria. This is an interesting and well written cross-sectional study which included 230 adults in Sokoto metropolis who received COVID-19 vaccine. The manuscript is an interesting and well written. The authors showed that up to 66.3% of the adverse reactions were mild, 15.2% of the respondents had severe reactions of which 22.7% were admitted to a health facility. The development of AEFI was linked to the presence of an underlying medical condition, a previous history of AEFI, and a history of drug reaction. The paper seems well organized.

Specific Comments: It would be helpful for the authors to respond to the following observation and revise the manuscript.

1. The title spells ‘COVID-19’ as ‘COVID 19’ whereas the remaining body it spells as ‘COVID-19’. WHO used Coronavirus disease as COVID-19. So the title should be revised.

2. Abstract: The last sentence of the Result section says that the development of AEFI was linked to the absence of an underlying medical condition………….. I think the it is better to interpret the result as ‘The development of AEFI was linked to the presence of an underlying medical condition……..’

3. Materials and Method: The duration/ time period of conducting the survey, inclusion or exclusion criteria should be include.

4. Results:

A) The author mentioned that 230 participants were included in the study. In Table 1, the sum of the Age sub-group shows 231.

B) Figure 1: The total number of respondents is shown as 231.

C) Figure 2: Y axis needs label.

D) Table 2: Number of AEFIs experienced respondents is shown 184 (n). There no explanation why this number decreased from 230 to 184.

E) Table 3: Under In the middle column, the total respondents equals 231 (instead of 230). In the right most column (Test Statistic) the box is left empty.

5. The authors have designated the AEFI as mild, moderate and severe. But they did not define the basis of this classification.

**Do you want your identity to be public for this peer review?** For information about this choice, including consent withdrawal, please see our Privacy Policy

Reviewer #1: **Yes: ** Md. Mustafizur Rahman

---

## [Author Response · Author response to Decision Letter 1]

30 Apr 2023

All issues raised by the reviewer(s) have been addressed by the authors; you will find the summary of our responses in the file titled "Response to Reviewer"

---

## [Decision Letter · Decision Letter 1]

15 May 2023

Dear Dr. Adamu,

Thank you for submitting your manuscript to PLOS ONE. After careful consideration, we feel that it has merit but does not fully meet PLOS ONE’s publication criteria as it currently stands. Therefore, we invite you to submit a revised version of the manuscript that addresses the points raised during the review process.

We look forward to receiving your revised manuscript.

Kind regards,

Mathumalar Loganathan Fahrni

Academic Editor

PLOS ONE

Additional Editor Comments:

Dear authors, the current manuscript requires several edits, which I hope you can complete and are then able to fulfil all the requirements.

1-Please provide study limitation, study strength, study implication and future recommendation as headings. The given background is inclusive. Please provide the hypothesis and problem statement.

2-Please include the rationale to study.Please organize the methodology and include headings such as eligibility criteria, inclusion and exclusion criteria.

3-Please explain the study design in detail.

4-As the study is cross-sectional, kindly provide a Strobe statement and strobe checklist.

5-Why was the GPower 3.1 tool used to determine the sample size? What were the advantages compared to Power and Sample Size Calculation (PASS) or OpenEpi? Justify the method of calculation. The total population of those who received COVID-19 vaccine at the time of the study was less than 10,000- please justify this figure and the relatively small sample size requirement estimate.

6- Rephrase the estimate of 90% and the statement "Using the above two formulae, sample size of 207 participants was estimated. A response rate of 90% was anticipated, thus the final sample size (nf) was adjusted to 230, by dividing the calculated sample size (207) by the anticipated response rate (0.9); therefore, 230 study participants were enrolled into the study". Provide references.

7-Please include questionnaire administration as a subheading and support the pre-test statements with references. Please provide provide the questionnaire as supplementary file.

8- Please explain and rewrite clearly the following statement, "The questionnaire was pretested among adult men and women who received COVID-19 vaccines from PHC Yar Akija. Necessary amendments were made thereafter, and the instrument was found valid. Personnel used for data collection comprised the researchers with the help of three research assistants. Data collection was done using the questionnaire described above. The questionnaire was designed and deployed to a web-based account created on https://kobo.humanitarianresponse.info. The deployed questionnaire was then accessed by downloading Open Data Kit (ODK) app on Android devices and was used for data collection. "

Please elaborate on the tool used, its benefits (web based?), and provide information on PHC Yar Akija's population demographics.

9-Please add the heading, "statistical analyses". The current provided limitation does not address all bias. Please provide in detail sampling bias, selection bias, recall bias, measurement bias. Please provide timeline of data collection.

10- Rewrite sections of the discussion and compare with findings of available literature.

Reviewers' comments:

Reviewer's Responses to Questions

**Comments to the Author**

Reviewer #1: All comments have been addressed

Reviewer #2: (No Response)

2. Is the manuscript technically sound, and do the data support the conclusions?

Reviewer #1: Yes

Reviewer #2: No

3. Has the statistical analysis been performed appropriately and rigorously?

Reviewer #1: Yes

Reviewer #2: No

4. Have the authors made all data underlying the findings in their manuscript fully available?

Reviewer #1: Yes

Reviewer #2: No

5. Is the manuscript presented in an intelligible fashion and written in standard English?

Reviewer #1: Yes

Reviewer #2: No

Reviewer #1: Initial recommendation to the Editor: The manuscript can be accepted with minor revision as shown in the following comments:

To the authors:

Thank you for properly addressing the previous queries. Here, during careful checking the manuscript I have found some more areas to be clarified by the authors. I guess you would find these as minor revision.

#1 Figure 1. Y-axis is missing label. Even the values are inserted, it is better to add y-axis label.

#2, Page 11, “N= Total population of adults who received COVID-19 from the 3 selected health facilities (HFs) = 1300”

I think instead of ‘COVID-19’ it will be ‘COVID-19 vaccine’

#3, a) Table 3: In the last column, the symbol is missing, probably. It would be ‘chi-square’. Pease ensure the symbol is typed correctly without any missing.

b) It is not clearly mentioned in the Table by which statistical parameters were determined to estimate the association. Please confirm whether you used ‘OR’ or ‘Chi square’. Then confirm the findings of statistical analysis of the data shown in this Table 3.

Reviewer #2: 1-Please organize the methodology and include heading such as eligibility criteria, inclusion criteria and exclusion criteria.

2-Please provide heading of study design and explain in details.

3-As the study is cross-sectional, kindly provide a strobe statement and strobe checklist.

4-Why you did not use GPower 3.1 tool to determine the sample size? What about Power and Sample Size Calculation (PASS) or OpenEpi? Justify your manual method of calculation. how you know that total population of those who received COVID-19 vaccine at the time of the study was less than 10,000? Please give heading of sample size and check your calculations and formulas and explain and justify it. Your sample size is too small and need to justify it.

5- There is lot of confusion in your sample size caluclation why did you estimate 90% as per your statement "Using the above two formulae, sample size of 207 participants was estimated. A response rate of

90% was anticipated, thus the final sample size (nf) was adjusted to 230, by dividing the calculated

sample size (207) by the anticipated response rate (0.9); therefore, 230 study participants were

enrolled into the study"

Why did you estimate it and why did you adjust it? Please justify rational and provide some reference on your methodology. Please provide with minimum 3-4 reference on rational about estimation of 90%? Did you estimate before the study starts and why 90%? its a high number. Please justify in details with reference.

6-Please include heading of questionnaire and even though its a pretest, there must be a literature support. Please explain and provide details. Please provide provide the questionnaire as supplementary file.

7- please explain and write clear the following statement "The questionnaire was pretested among adult men and women who received COVID-19 vaccines

from PHC Yar Akija. Necessary amendments were made thereafter, and the instrument was found

valid.

Personnel used for data collection comprised the researchers with the help of three research

assistants.

Data collection was done using the questionnaire described above. The questionnaire was designed

and deployed to a web-based account created on https://kobo.humanitarianresponse.info. The

deployed questionnaire was then accessed by downloading Open Data Kit (ODK) app on Android

devices and was used for data collection. "

Please elaborate the tool used and what are the benefits of web based accounts? There is still no information on PHC Yar Akija. It is confusing.

Please add the heading of statistical analysis and explain it in the methodology.

Please provide study limitation, study strength, study implication and future recommendation as heading and explain.

The given background is inclusive. Please provide testing hypothesis. Please include problem background and explain in details the problem.

Please include heading of theoretical background and provide rational to your study.

The current provided limitation does not address all bias. Please provide in detail sampling bias, selection bias, recall bias, measurement bias.

Please provide timeline of data collection.

I have read your discussion and your discussion is weak and does not compare your finding with provide available scientific literature. Rewrite your discussion and include citation with updated literature using data from countries of other parts of world. Please write your discussion according to given objective of your study.

**Do you want your identity to be public for this peer review?** For information about this choice, including consent withdrawal, please see our Privacy Policy

Reviewer #1: **Yes: ** Dr. Md. Mustafizur Rahman

Reviewer #2: **Yes: ** Muhammad Shahzad Aslam

---

## [Author Response · Author response to Decision Letter 2]

16 Jul 2023

1. Need for statistical significance to support result findings

1. Response: Respective p-values inserted

2. Contradictory statements in the result and conclusion

2. Response: Contradiction rectified; it is supposed to be presence of underlying disease not absence of…

1. Background, problem statement, rationale and hypothesis should be provided

Response: Done

1. To organize the methodology and include headings such as eligibility criteria, inclusion and exclusion criteria.

1. Response: Subheadings included

2. Please explain the study design in detail.

4-As the study is cross-sectional, kindly provide a Strobe statement and strobe checklist.

2. Response: Subheadings included

3. Why was the GPower 3.1 tool used to determine the sample size? What were the advantages compared to Power and Sample Size Calculation (PASS) or OpenEpi? Justify the method of calculation.

3. Response: Sample size was calculated manually using the Cochran formula; detailed calculation has been given and the formulae used were well referenced, including the modified Cochran formula which is used in calculating sample size when the size of the target population is not so large (<10,000); this is an acceptable method of calculating size.

4. The total population of those who received COVID-19 vaccine at the time of the study was less than 10,000- please justify this figure and the relatively small sample size requirement estimate.

4. Response: In Nigeria, there is a COVID-19 register at each COVID-19 vaccination site where records of all those who received the vaccine are kept; the total number of the recipients in the register was not up to 10,000 (i.e. in the three health facilities selected for the study. This is not surprising considering the fact that vaccine hesitancy has been an issue in Nigeria even before COVID-19 outbreak; this perhaps explains the relatively small number of those vaccinated. healDuration/time period indicated. The inclusion/exclusion have been stated already; stated in the 3rd paragraph of material and method section

5. Rephrase the estimate of 90% and the statement "Using the above two formulae, sample size of 207 participants was estimated. A response rate of 90% was anticipated, thus the final sample size (nf) was adjusted to 230, by dividing the calculated sample size (207) by the anticipated response rate (0.9); therefore, 230 study participants were enrolled into the study".

Provide references. Why did you estimate it and why did you adjust it? Please justify rational and provide some reference on your methodology. Please provide with minimum 3-4 reference on rational about estimation of 90%? Did you estimate before the study starts and why 90%? its a high number. Please justify in details with reference

5. Response: Statement rephrased, detailed explanation given and reference provided

6. Please include questionnaire administration as a subheading and support the pre-test statements with references. Please provide provide the questionnaire as supplementary file.

6. Response: Done

7. Please explain and rewrite clearly the following statement, "The questionnaire was pretested among adult men and women who received COVID-19 vaccines from PHC Yar Akija. Necessary amendments were made thereafter, and the instrument was found valid. Personnel used for data collection comprised the researchers with the help of three research assistants. Data collection was done using the questionnaire described above. The questionnaire was designed and deployed to a web-based account created on https://kobo.humanitarianresponse.info. The deployed questionnaire was then accessed by downloading Open Data Kit (ODK) app on Android devices and was used for data collection. "

7. Response: Explanation given

8. Please elaborate on the tool used, its benefits (web based?), and provide information on PHC Yar Akija's population demographics.

8. Response: Benefits of using ODK App has been given and additional information on PHC Yar Akija has also been provided

9. Please add the heading, "statistical analyses". The current provided limitation does not address all bias. Please provide in detail sampling bias, selection bias, recall bias, measurement bias. Please provide timeline of data collection.

9. Response: Detail on handling bias is provided (see study strength

11. Figure 1. Y-axis is missing label. Even the values are inserted, it is better to add y-axis label.

11. Response: Y-axis label added

12. Page 11, “N= Total population of adults who received COVID-19 from the 3 selected health facilities (HFs) = 1300”

I think instead of ‘COVID-19’ it will be ‘COVID-19 vaccine’

12. Response: Corrected; vaccine added

13. a) Table 3: In the last column, the symbol is missing, probably. It would be ‘chi-square’. Pease ensure the symbol is typed correctly without any missing.

b) It is not clearly mentioned in the Table by which statistical parameters were determined to estimate the association. Please confirm whether you used ‘OR’ or ‘Chi square’. Then confirm the findings of statistical analysis of the data shown in this Table 3

13. Response: Test statistic (Pearson chi square) defined below the table; Pearson chi square test was used to test for association (with the respective p-values)

14. Rewrite sections of the discussion and compare with findings of available literature

14. Response: All reported findings have been compared with other findings; implications of findings also discussed.

Additional references have also been given

---

## [Editor Report · Decision Letter 2]

8 Aug 2023

Dear Dr. Habibullah Adamu,

Thank you for submitting your manuscript to PLOS ONE. After careful consideration, we feel that it has merit but does not fully meet PLOS ONE’s publication criteria as it currently stands. Therefore, we invite you to submit a revised version of the manuscript that addresses the points raised during the review process.

Please refer to attached document.

We look forward to receiving your revised manuscript.

Kind regards,

Mathumalar Loganathan Fahrni

Academic Editor

PLOS ONE
---

## [Author Response · Author response to Decision Letter 3]

12 Feb 2024

Title

1. Suggested adding “an” to adult population We thought adding “an” to it will suggest we are referring to a particular adult population in Sokoto state; however, our study did not focus on any specific/particular adult population, it considered all eligible adults who received COVID 19 vaccine from the selected health facilities

A Abstract

1. "As all vaccines are associated with some adverse reactions (ARs)" - to be removed from objectives.

2. Methods - COVID-19 vaccine - "s" to be added

3. Results: "M"ajority

4. Conclusion: Remove comma before ","with body weakness

5. 1. Done

2. Done

3. Done

4. Done

B Introduction

1. Corrections on lines 1 and 2

2. To consider removing paragraphs 2 and 3

3. To remove the phrases “problem statement”, “rationale”, hypothesis”

4. 1. Done

2. Paragraph 3 removed

3. The phrases were not there in the initial version of the manuscript, but the authors were advised to include it in the previous review; the phrases have now been removed

C Materials and method

1. Move around the contents so they fall under the specific subheadings. Use subheadings "Study design and sampling", "Study instruments", Inclusion and exclusion criteria", "Enrollment"

2. Please provide as supplementary material a completed checklist stating page numbers where each of the criterion was fulfilled https://www.strobe-statement.org/checklists/

3. Rephrase so it reflects the past actions of the research team involved. "So if a researcher anticipates that up to 90% of the subjects will complete the questionnaires, he should divide the sample size by 90% (i.e. 0.9)."

4. The content under data collection subsection to be reorganised and made clearer. Currently unclear who made the calls, as there was a mention of research assistants in the preceding para.

5. Rephrase, "confidentiality of the information provided by the participants was also assured."

6. Study limitations to be moved to end of discussion section. 1. The contents as presented herein are based on earlier corrections/suggestions by reviewers of the manuscript

2. Study designed explained;

3. In the previous review of the manuscript, the authors were asked to explain (with reference) their reason for adjusting the sample size to cater for non-response of the study participants

The past actions of the researchers are stated in the paragraph that followed the explanation

4. Done

5. Rephrased

6. Moved

D Result

1. Please recalculate response rate using initial number eligible for inclusion, then excluded due to missing phone numbers, approached for interview but did not give consent etc

2. Figure 1 can be removed as it is described in text.

3. Rephrase "On bivariate analysis using Pearson chi square test

4. Logistic regression findings: both presence of comorbidities and hx of adverse drug rx have very wide CIs. Please explain the steps taken when performing the regression analysis.

1. The initial number eligible for inclusion is 230 (based on the sample size calculated) and all those approached consented and responded to the questions. The only explanation was that, some study participants rescheduled in interview to a more convenient time for them.

2. As the content of the chart is among the key objectives of the study, the researchers felt it would be better to present it in a chart because readers could miss it if presented in text only

3. Rephrased

4. The dependent variable was experience of AEFI following COVID 19 vaccination (coded as No-0 and Yes – 1). Three factors were found to have significant association with experience of AEFI, thus they were selected into the covariate box (after selecting regression in SPSS) as the independent variables (IV); experience of AEFI was selected into the Dependent variable box. Since all the 3 IV are categorical (with Yes/No options), they were further selected into the “Categorical Covariates” box with the last option (“No”) as the reference category. 95% CI was selected and the method used was “Enter”

E Discussion

1. Please state the corresponding figures obtained in at least one other country.

"Similar studies conducted in Europe,18,26 Asia,15 and Africa16,19 also reported a high prevalence of adverse reactions to COVID-19”

2. Any major reaction reported at all? How does the state deal with major, if any? Eg. yellow reporting form, pharmacovigilance processes

3. Study limitation - if only one type then mention consistently throughout, objective, methods, results, etc. Consider adding to title too.

One major limitation of this study is the fact that all the respondents received only the AstraZeneca vaccine, thus it was not possible to compare the rate of adverse events across all the vaccines approved for the COVID-19 vaccination 1. Done (a study from Mexico added)

2. Explanation given

3. Done

F Conclusion/recommendation

Remove all the subheadings after conclusion. The contents could be organised as a separate para in the discussion (after strengths and limitation)

Done

G Reference list

Nil

---

## [Decision Letter · Decision Letter 3]

22 May 2024

Dear Dr. Adamu,

Thank you for submitting your manuscript to PLOS ONE. After careful consideration, we feel that it has merit but does not fully meet PLOS ONE’s publication criteria as it currently stands. Therefore, we invite you to submit a revised version of the manuscript that addresses the points raised during the review process.

We look forward to receiving your revised manuscript.

Kind regards,

Francesco Sessa, Ph.D., MS

Academic Editor

PLOS ONE

Additional Editor Comments:

The reviewers raised several points that should be improved before publishing the manuscript. Please, provide a point-by-point rebuttal letter.

Reviewers' comments:

Reviewer's Responses to Questions

**Comments to the Author**

Reviewer #3: All comments have been addressed

Reviewer #4: (No Response)

2. Is the manuscript technically sound, and do the data support the conclusions?

Reviewer #3: Yes

Reviewer #4: (No Response)

3. Has the statistical analysis been performed appropriately and rigorously?

Reviewer #3: Yes

Reviewer #4: (No Response)

4. Have the authors made all data underlying the findings in their manuscript fully available?

Reviewer #3: Yes

Reviewer #4: (No Response)

5. Is the manuscript presented in an intelligible fashion and written in standard English?

Reviewer #3: Yes

Reviewer #4: (No Response)

Reviewer #3: I read the article carefully, I think it is interesting and useful also for emerging countries.

However, I believe it needs a greater international scope.

- a benchmark should be done with other similar studies published in other countries

- the previous suggestion would help to expand the bibliography which appears very limited

Reviewer #4: The introduction Lacks a clear statement of the study's hypothesis and specific aim. The authors should explicitly state their hypothesis and the primary objective of their research. For instance, the hypothesis could be: "The study hypothesizes that a significant proportion of COVID-19 vaccine recipients in Sokoto Metropolis experience adverse events following immunization (AEFI), with specific demographic and clinical factors influencing these events." The aim should be clearly articulated, such as: "The aim of this study is to determine the prevalence and pattern of AEFI following COVID-19 vaccination among adults in Sokoto Metropolis and to identify factors associated with these adverse events."

The authors should detail the inclusion and exclusion criteria for the sample more comprehensively. For example, specify the exact criteria for inclusion (e.g., age range, health status) and exclusion (e.g., specific medical conditions, previous severe reactions to vaccines).

The term "adverse events following immunization (AEFI)" is mentioned but not clearly defined in the methods section. The authors should provide a precise definition and examples of what constitutes an AEFI in the context of this study.

The timeline for data collection is not specified. The authors should include the start and end dates of data collection to provide context for the findings and to account for any changes in vaccination policies or external factors during the study period.

The authors claim that their data is representative of the entire population, but this assertion needs justification. They should explain why the chosen sample and methods are representative of the broader population of Sokoto Metropolis, considering demographic and geographic diversity.

Discuss the possibility of sampling bias due to the selection of participants from specific health facilities and the use of phone interviews, which may exclude those without access to phones or those who are less willing to participate.

The authors should consider the impact of their sampling method on the generalizability of the findings. For example, patients from different socioeconomic backgrounds or those who received vaccinations at different times or locations might have different experiences.

Since the data rely on self-reported information collected via phone calls, there is a risk of recall bias. Participants might not accurately remember or report their experiences, especially if significant time has passed since their vaccination.

There might be inconsistencies in how adverse events are reported and recorded. The authors should discuss any steps taken to ensure the accuracy and consistency of the data collected through the questionnaires.

The conclusion summarizes the main findings effectively. However, it should reiterate the importance of addressing the study's limitations and suggest directions for future research

**Do you want your identity to be public for this peer review?** For information about this choice, including consent withdrawal, please see our Privacy Policy

Reviewer #3: **Yes: ** Matteo Bolcato

Reviewer #4: No

---

## [Author Response · Author response to Decision Letter 4]

14 Jul 2024

Introduction

1. Authors should explicitly state the hypothesis and primary objective of the study

1. Done

Materials and method

1. The authors should detail the inclusion and exclusion criteria for the sample more comprehensively. For example, specify the exact criteria for inclusion (e.g., age range, health status) and exclusion (e.g., specific medical conditions, previous severe reactions to vaccines).

2. The authors should provide a precise definition and examples of what constitutes an AEFI in the context of this study.

3. The timeline for data collection is not specified

4. The authors claim that their data is representative of the entire population, but this assertion needs justification. They should explain why the chosen sample and methods are representative of the broader population of Sokoto Metropolis, considering demographic and geographic diversity. Authors should also discuss the possibility of sampling bias due to the selection of participants from specific health facilities and the use of phone interviews, which may exclude those without access to phones or those who are less willing to participate.

5. Since the data rely on self-reported information collected via phone calls, there is a risk of recall bias. Participants might not accurately remember or report their experiences, especially if significant time has passed since their vaccination.

1. Done

2. Done

3. The timeline has already been specified in the manuscript (page 9); see the last sentence under the subsection on questionnaire administration

4. At the time of collecting the data for the study, very few health facilities were providing COVID 19 vaccination service. Therefore, all recipients of the vaccine within the metropolis must access the service from one of these health facilities irrespective of their gender, race/ethnicity, religion or social status; thus, the sociodemographic diversity within the metropolis is taken into account. The only tertiary hospital within the metropolis was the one selected for this study; likewise, the primary and secondary health facilities selected in this study were selected from the few remaining health facilities providing the vaccination service via a probability sampling technique. All categories of health facilities (primary, secondary and tertiary) providing COVID 19 vaccination services at the time were included in the sampling frame. Based on the recipients’ record in the COVID 19 registers at the selected HFs, nearly all the recipients had their phone numbers of their next of kin’s number recorded.

5. This was mentioned as part of the limitations of the study, nevertheless, we believe their recall was good enough since all the adverse reactions were mentioned to the respondents during the phone call interviews.

6.

Results

Discussion

6. Authors should reiterate the importance of addressing the study's limitations and suggest directions for future research

7. Done

---

## [Decision Letter · Decision Letter 4]

10 Sep 2024

Dear Dr. Adamu,

Thank you for submitting your manuscript to PLOS ONE. After careful consideration, we feel that it has merit but does not fully meet PLOS ONE’s publication criteria as it currently stands. Therefore, we invite you to submit a revised version of the manuscript that addresses the points raised during the review process.

**ACADEMIC EDITOR:** After several rounds of revision, the manuscript has substantially improved. I believe that the paper is suitable for publication, pending the minor modifications suggested by Reviewer #3 and improvements to the reference section. The authors should consider adding the following important references: DOI: 10.3390/diagnostics11060955; DOI: 10.3389/fphar.2024.1308768; DOI: 10.1038/s41467-022-35653-z; DOI: 10.3390/jcm10245876. 

We look forward to receiving your revised manuscript.

Kind regards,

Francesco Sessa, Ph.D., MS

Academic Editor

PLOS ONE

Journal Requirements:

Additional Editor Comments:

After several rounds of revision, the manuscript has substantially improved. I believe that the paper is suitable for publication, pending the minor modifications suggested by Reviewer #3 and improvements to the reference section. The authors should consider adding the following important references: DOI: 10.3390/diagnostics11060955; DOI: 10.3389/fphar.2024.1308768; DOI: 10.1038/s41467-022-35653-z; DOI: 10.3390/jcm10245876.

Reviewers' comments:

Reviewer's Responses to Questions

**Comments to the Author**

Reviewer #3: All comments have been addressed

2. Is the manuscript technically sound, and do the data support the conclusions?

Reviewer #3: Yes

3. Has the statistical analysis been performed appropriately and rigorously?

Reviewer #3: Yes

4. Have the authors made all data underlying the findings in their manuscript fully available?

Reviewer #3: Yes

5. Is the manuscript presented in an intelligible fashion and written in standard English?

Reviewer #3: Yes

Reviewer #3: I have carefully read the article and believe it deserves publication. Especially after several suggestions from the reviewers, the text appears adequate and the method used robust.

I believe it could be further improved by making a brief mention of how the vaccine studied can help achieve equity of access doi: 10.3390/vaccines9060538.

It would also be important, as already indicated in the conclusions, to indicate how the information to be given to the patient is useful for compliance and fairness.

**Do you want your identity to be public for this peer review?** For information about this choice, including consent withdrawal, please see our Privacy Policy

Reviewer #3: **Yes: ** Matteo Bolcato

---

## [Author Response · Author response to Decision Letter 5]

25 Nov 2024

SN COMMENTS/OBSERVATIONS OF REVIEWER ACTION TAKEN

Title

A Abstract

Nil comment from reviewer

B Introduction

Nil

C Materials and method

Nil

D Results

Nil

E Discussion

It would also be important, as already indicated in the conclusions, to indicate how the information to be given to the patient is useful for compliance and fairness

Response - Done

F Conclusion/recommendation

Nil

G Reference list

Nil

Supporting document

---

## [Decision Letter · Decision Letter 5]

20 Dec 2024

Prevalence and pattern of adverse events following COVID-19 vaccination among adult population in Sokoto metropolis, northwest, Nigeria

PONE-D-22-29819R5

Dear Dr. Adamu,

We’re pleased to inform you that your manuscript has been judged scientifically suitable for publication and will be formally accepted for publication once it meets all outstanding technical requirements.

Kind regards,

Francesco Sessa, Ph.D., MS

Academic Editor

PLOS ONE

Additional Editor Comments (optional):

Following a thorough and productive revision process, the manuscript is now ready to be published.

Reviewers' comments:

Reviewer's Responses to Questions

**Comments to the Author**

Reviewer #3: All comments have been addressed

2. Is the manuscript technically sound, and do the data support the conclusions?

Reviewer #3: Yes

3. Has the statistical analysis been performed appropriately and rigorously?

Reviewer #3: Yes

4. Have the authors made all data underlying the findings in their manuscript fully available?

Reviewer #3: Yes

5. Is the manuscript presented in an intelligible fashion and written in standard English?

Reviewer #3: Yes

Reviewer #3: the authors have carefully followed the suggestions of the reviewers, in my opinion the text is now publishable

**Do you want your identity to be public for this peer review?** For information about this choice, including consent withdrawal, please see our Privacy Policy

Reviewer #3: **Yes: ** Matteo Bolcato

---

## [Editor Report · Acceptance letter]

PONE-D-22-29819R5

PLOS ONE

Dear Dr. Adamu,

I'm pleased to inform you that your manuscript has been deemed suitable for publication in PLOS ONE. Congratulations! Your manuscript is now being handed over to our production team.

Kind regards,

on behalf of

Lecturer Francesco Sessa

Academic Editor

PLOS ONE